# Anti-Inflammatory Effect of Beta-Caryophyllene Mediated by the Involvement of TRPV1, BDNF and trkB in the Rat Cerebral Cortex after Hypoperfusion/Reperfusion

**DOI:** 10.3390/ijms23073633

**Published:** 2022-03-26

**Authors:** Maria Pina Serra, Marianna Boi, Antonella Carta, Elisabetta Murru, Gianfranca Carta, Sebastiano Banni, Marina Quartu

**Affiliations:** Department of Biomedical Sciences, University of Cagliari, Cittadella Universitaria, 09042 Monserrato, Italy; mpserra@unica.it (M.P.S.); marianna.boi@unica.it (M.B.); a.carta86@studenti.unica.it (A.C.); m.elisabetta.murru@gmail.com (E.M.); giancarta@unica.it (G.C.); banni@unica.it (S.B.)

**Keywords:** beta-caryophyllene, neuroinflammation, acute bilateral common carotid artery occlusion, TRPV1, BDNF, trkB, GFAP, Iba1, Western blot, immunohistochemistry

## Abstract

We have previously shown that bilateral common carotid artery occlusion followed by reperfusion (BCCAO/R) is a model to study early hypoperfusion/reperfusion-induced changes in biomarkers of the tissue physiological response to oxidative stress and inflammation. Thus in this study, we investigate with immunochemical assays if a single dose of beta-caryophyllene (BCP), administered before the BCCAO/R, can modulate the TRPV1, BDNF, and trkB receptor in the brain cortex; the glial markers GFAP and Iba1 were also examined. Frontal and temporal-occipital cortical regions were analyzed in two groups of male rats, sham-operated and submitted to BCCAO/R. Six hours before surgery, one group was gavage fed a dose of BCP (40 mg/per rat in 300 μL of sunflower oil), the other was pre-treated with the vehicle alone. Western blot analysis showed that, in the frontal cortex of vehicle-treated rats, the BCCAO/R caused a TRPV1 decrease, an increment of trkB and GFAP, no change in BDNF and Iba1. The BCP treatment caused a decrease of BDNF and an increase of trkB levels in both sham and BCCAO/R conditions while inducing opposite changes in the case of TRPV1, whose levels became higher in BCCAO/R and lower in sham conditions. Present results highlight the role of BCP in modulating early events of the cerebral inflammation triggered by the BCCAO/R through the regulation of TRPV1 and the BDNF-trkB system.

## 1. Introduction

Accumulating evidence shows that inflammation plays a crucial role in the pathogenesis of cerebral ischemia and represents a target for promising therapeutic intervention [1,2]. While the severity and treatment of ischemia-induced functional and metabolic dysregulation depend on various factors [3,4,5]—the vessel occluded, permanent or temporary hypoperfusion, the timing of intervention—preventive strategies to reduce the risk factors for ischemic stroke, such as carotid artery stenosis, have extensively gained attention from researchers [6]. The knowledge accrued from the utilization of preclinical models of cerebral hypoperfusion/revascularization injury [5] has allowed not only to identify the nature and dynamics of the reperfusion injury [1,6,7,8] but also to correlate hypoperfusion/reperfusion to reliable biomarker changes. So far, evidence in this area of research shows that the acute transient bilateral common carotid artery occlusion followed by reperfusion (BCCAO/R) is a model that can mimic in vivo the biological effects of acute cerebral oxidative stress and the early formation of a deleterious pro-inflammatory milieu [9,10]. Indeed, evidence shows that the BCCAO/R affects the brain tissue physiological homeostasis as early as 30 min [11,12] after surgery and appears to directly correlate to molecular marker adaptive changes in brain tissue and plasma [10,11,12,13,14,15,16,17,18,19,20,21]. 

Emerging evidence reveals that natural products with anti-inflammatory and anti-oxidant properties exert preventive effects against the cerebral molecular dysfunction induced by acute BCCAO/R [12,15,18,19,20]. Among the most promising natural products, the bicyclic sesquiterpene beta-caryophyllene (BCP), a compound that is commonly found in the essential oils of many food plants [22,23], and represents a primary component in the *Cannabis sativa* L. plant [22]. BCP is a dietary phytocannabinoid exhibiting its therapeutic effects acting as a selective agonist for the CB2 cannabinoid receptor (CB2R) (Ki = 155 nM) [24,25] and the peroxisome proliferator activating receptor alpha (PPARalpha) [26,27]. BCP has various phytotherapeutic properties [23,28] that were shown widely in different animal models of metabolic and neurological diseases based on inflammatory states, such as formalin test [29], arthritis [30], carrageenan- and prostaglandin E1-induced edema [24,31], colitis [32], cisplatin-induced nephrotoxicity [33], obesity and dyslipidemia, hepatic steatosis, diabetes, cardiovascular disorders [27], focal cerebral ischemia [34], and transient BCCAO/R [19]. In particular, we recently characterized the effects of a single dose of BCP against the pro-inflammatory milieu induced by the BCCAO/R in the rat [19] and demonstrated that BCP affects the endocannabinoid system by increasing the tissue levels of 2-Arachidonoylglycerol (2-AG), N-arachidonoylethanolamide (AEA), Palmitoylethanolamide (PEA), Oleoylethanolamide (OEA), and the expression of CB receptors, increases the expression of PPARalpha receptor, spares the basal tissue levels of docosahexaenoic acid (DHA), decreases the plasmatic levels of AEA and reverses the increase of plasmatic lipoperoxides [19]. 

The transient receptor potential vanilloid type-1 (TRPV1) is a thermosensitive cation channel that was widely investigated as a pain receptor in primary sensory neurons [35,36,37,38,39]. TRPV1 is a polymodal receptor that can be activated by exogenous and endogenous stimuli, such as noxious heat, deviations from homeostatic pH, compounds such as capsaicin and resiniferatoxin, endocannabinoids and products of lipoxygenases [37,40,41,42,43]. From its functional role in nociception, including also its involvement in the efferent function of sensory nerve endings, leading to neuropeptide release and consequent neurogenic effects [37,43,44,45], the understanding of TRPV1 signaling in the modulation of inflammation in both peripheral organs and central nervous system (CNS) is continually evolving [44,46,47]. TRPV1 expression is promptly and strongly modulated by inflammatory mediators [46]. Though the actual regional distribution of TRPV1 in the CNS, apart from territories of primary sensory innervation, is still controversial [48], TRPV1 is expressed by neurons and glia [46,49,50]. The latter is thought to represent the link between TRPV1 and neuroinflammation since its activity is regulated by TRPV1 and, conversely, TRPV1 sensitization is triggered upon activation by pro-inflammatory agents [46,47,50]. Despite the still incomplete evidence regarding the circumstances and timing when TRPV1 may signal either pro-inflammatory or anti-inflammatory effects, TRPV1 activation appears to play a role in the severity of effects on cerebral tissue induced by hypoxic ischemia [51] and ischemia/reperfusion [52]. In this context, it is relevant that, for its ability to be activated by AEA, AEA analogs, and PEA, TRPV1 has also been identified as the ionotropic cannabinoid receptor, thus establishing a further connection with its anti-inflammatory effects [53]. Furthermore, studies on nociception have also indicated a role of TRPV1 in neuroplasticity and shown that TRPV1 could induce long-term potentiation (LTP) via brain-derived neurotrophic factor (BDNF) [54]. The BDNF, in turn, regulates TRPV1 expression and α-amino-3-hydroxy-5-methyl-4-isoxazole propionic acid (AMPA) receptors, which have a role in inducing the generation of action potentials in the postsynaptic element [54].

Therefore, given that revascularization of common carotid arteries is likely to induce a pro-inflammatory milieu at the end of the hypoperfusion process, here we extend our previous observations on the neuropreventive effects of an acute dose of BCP on the molecular changes induced by the BCCAO/R in the rat. With this aim, utilizing Western blot and immunohistochemical assays, we examined the levels of TRPV1, BDNF, and its high affinity tropomyosin receptor kinase B (trkB), as well as those of the glial fibrillary acidic protein (GFAP) and ionized calcium-binding adapter molecule 1 (Iba1). The investigated regions were the frontal cortex, the brain region that is affected by the hypoperfusion/reperfusion due to the transient occlusion of the common carotid arteries [16], and in the temporal-occipital cortex, an area supposedly not influenced by the BCCAO/R [16]. Results are discussed envisaging the use the phytocannabinoid BCP as a dietary supplement to prevent the changes in the levels of TRPV1, BDNF, and trkB as putative molecular markers of adaptive brain tissue plasticity in response to the pro-inflammatory state induced by the cerebral hypoperfusion/reperfusion. 

## 2. Results

### 2.1. Western Blot Assays

The effects of the BCCAO/R without and with preventive administration of BCP on the relative levels of TRPV1, BDNF, trkB, GFAP, and Iba1 proteins are summarized in Table 1 and shown in Figure 1 and Figure 2. Statistical analysis of O.D. values of the immunostained protein bands performed by two-way ANOVA (main factors BCCAO/R and BCP) (Table 1) revealed that both the BCCAO/R-induced molecular changes and the effect of the BCP pre-treatment occurred in the frontal cortex, whereas no statistically significant differences were evident in the temporal-occipital cortex.

#### 2.1.1. The TRPV1 Protein Levels

The antibody against TRPV1 labeled a protein band with a relative mw of 89.9 kDa (Figure 1 Left), consistent with the reported mw of the receptor monomeric form [55]. Assessment of the TRPV1 densitometric values by a two-way ANOVA (Table 1) revealed an effect of BCP (*p* = 0.0072) and a significant BCP × BCCAO/R interaction (*p* < 0.0001). Pair-wise contrasts further showed that, in the vehicle-treated animals, a statistically significant decrease of TRPV1 relative protein levels, amounting to −201%, occurred in BCCAO/R vs. sham-operated rats (*p* < 0.0001). After the BCP pre-treatment, relative levels of the TRPV1 decreased in the sham-operated (−62%; *p* < 0.0001) while increased in the BCCAO/R rat group (+69%; *p* < 0.0001) as compared to the vehicle-treated rats; finally, a statistically significant increase of TRPV1 relative levels occurred in BCCAO/R-rats as compared to sham-operated ones (+42%; *p* < 0.0001).

#### 2.1.2. The BDNF Protein Levels

The antibody against BDNF labeled a protein band with a relative mw of about 13 kDa (Figure 1), according to the expected mw of the monomeric form of the protein [56,57]. Assessment of the BDNF densitometric values by a two-way ANOVA (Table 1) revealed an effect of BCP (*p* < 0.0001). The slight reduction of BDNF relative levels in the BCCAO/R rats (0.7592 ± 0.01 in sham rats vs. 0.6936 ± 0.004 in BCCAO/R rats) is consistent with previously published data [20]. Such a decrease was highly statistically significant with an unpaired *t*-test analysis (*p* < 0.0001), however, it did not reach the statistical significance with the pair-wise contrasts run after the ANOVA. Pair-wise contrasts instead showed that, after the BCP pre-treatment, the relative levels of BDNF protein were decreased in both sham-operated (−71%; *p* < 0.0001) and BCCAO/R rats (−42%; *p* = 0.0001) as compared to the vehicle-treated rats. 

#### 2.1.3. The trkB Protein Levels

The anti-trkB antibody, raised against the full-length isoform of the receptor protein, recognized a protein band with a relative mw of about 140 kDa (Figure 1), consistent with the reported mw of the receptor protein [57,58]. Assessment of the trkB densitometric values by a two-way ANOVA (Table 1) showed effects of the BCCAO/R (*p* < 0.0001) and of the BCP pre-treatment (*p* < 0.0001). Pair-wise contrasts revealed that, as a general rule, the BCP-treatment induced an increase of trkB relative levels amounting to about 97% (*p* = 0.0024) and 33% (*p* = 0.0028) in the sham-operated and BCCAO/R rats, respectively, vs. the vehicle-treated ones. In particular, in vehicle-treated animals, the trkB protein relative levels increased by 119% in BCCAO/R (*p* < 0.0001) vs. the sham-operated rats, while in the BCP-treated rats increased by 47% (*p* = 0.0002) in the BCCAO/R vs. the sham-operated. 

#### 2.1.4. The GFAP Protein Levels

The antibody against GFAP, as expected [59], recognized a protein band with a relative mw of about 50 kDa (Figure 2). Assessment of the GFAP densitometric values by a two-way ANOVA (Table 1) revealed an effect of BCP treatment (*p* < 0.0001) and a BCP × BCCAO/R interaction (*p* = 0.0006). Further post-hoc tests showed that, in vehicle-treated animals, the GFAP protein relative levels increased by 27% in BCCAO/R rats as compared to sham-operated ones (*p* = 0.0015). In BCCAO/R rats the GFAP relative levels decreased by 44% in BCP-treated compared to vehicle-treated rats (*p* < 0.0001).

#### 2.1.5. The Iba1 Protein Levels

The antibody against Iba1 labeled a protein band with a relative mw of about 17 kDa (Figure 2), according to the expected mw of the protein [60]. Assessment of the Iba1 densitometric values by a two-way ANOVA (Table 1) revealed an effect of BCCAO/R (*p* < 0.0007). 

### 2.2. Immunohistochemistry

To draw a parallel of the molecular changes observed by Western blot analysis with the tissue morphology, we performed immunostainings of the rat brain sections with the same antibodies used for Western Blot analysis. Immunoreactivities for all examined markers were localized to neuronal structures distributed throughout the rostro-caudal extension of the brain (Figure 3, Figure 4, Figure 5 and Figure 6). As regards the frontal cortex, TRPV1-LI appeared as sparse dot-like elements and tiny varicose nerve fibers distributed mostly close to blood vessels or around them (Figure 3 and Figure 6A,D,G). Neuronal perikarya showing a faint intracytoplasmic staining with a dust-like aspect were further detectable (Figure 6A,D). Labeling of the meningeal lining of the cortex was also detectable. By contrast with the scarcity of TRPV1-LI, the BDNF- and trkB-like immunoreactive structures were numerous and identifiable as neuronal perikarya and proximal processes and nerve fibers distributed throughout the cortical layers, having the aspect of loose networks of thin filaments and punctate elements in the superficial layers, and straight neuronal processes with a prevalent radial orientation in the deep layers (Figure 4 and Figure 5). Double labeling for either TRPV1 and GFAP (astrocyte marker) or TRPV1 and Iba1 (microglia marker), carried out by means of indirect immunofluorescence in selected series of brain sections from BCCAO/R rats showed that, in both BCP- and vehicle-treated rats, rare TRPV1-labelled nerve fibers and dot-like elements were also GFAP-immunoreactive (Figure 6C,F), whereas TRPV1/Iba1 colocalization was virtually absent (Figure 6B).

## 3. Discussion

The main finding of this study is that the BCP treatment in a single dose exerts preventive effects against the molecular changes induced by the BCCAO/R in the rat frontal cortex. In particular, the BCP-treatment (a) decreases the TRPV1 relative levels in sham rats while increasing them in the BCCAO/R condition; (b) decreases the relative levels of BDNF while inducing a concomitant increase of the trkB levels in both sham and BCCAO/R rats; (d) avoids changing of GFAP protein levels after the BCCAO/R. Additionally, present data enrich the knowledge of molecular markers whose expression is affected by early changes in response to a BCCAO/R protocol with the same length of hypoperfusion/reperfusion used in this study [16,18,19,20]. Thus, besides the previously reported effects on BDNF and trkB [20], present data support evidence that the BCCAO/R itself causes a sufficient tissue challenge to induce a significant decrease of TRPV1 and GFAP expression levels as compared to sham-operated rats. 

The transient BCCAO model is the most employed model of global cerebral hypoperfusion that can be adapted to investigate the efficacy of therapies in acute vascular accidents when the revascularization leads to the pathophysiological sequelae of reperfusion [5,61]. The BCCAO model, despite the blood flow compensation due to the circle of Willis, has been reported in the rat to cause a reduction of cerebral blood flow of 25 to 87% (depending on the brain region examined), amounting to about 63% in the frontal cortex to about 20% in the hippocampus vs. sham animals after 2.5 h of arterial ligation [13]. In our hands, by occluding the CCA with atraumatic miniclamps for 30 min and allowing reperfusion for 1 h, the BCCAO/R triggered several molecular changes that can be considered as a prodrome of cerebral oxidative stress and neuroinflammation [15,16,18,19,20]. Thus, in the cerebral tissue, changes comprise a reduced concentration of lipid profile components, such as the DHA, a structural membrane polyunsaturated fatty acid (PUFA); the accumulation of endocannabinoids and PEA; the increase of lipoperoxide concentration; the increase of expression of the enzyme cyclooxygenase-2 (COX-2), and CB1 and CB2 receptors [16,18,19]. The detectability of some lipid components, such as the lipoperoxides, in the plasma further supports the view that the BCCAO/R affects molecules whose presence in the circulation is possibly correlatable to the imbalance of tissue homeostasis induced by the transient vascular occlusion. 

### 3.1. TRPV1 and N-Acyl Amides in BCCAO/R with and without BCP

Regarding the TRPV1, its expression changes following the BCCAO/R were not unexpected due to converging literature evidence on its polymodal activation by different agents and our previous findings on the molecular changes induced by BCCAO/R [16]. 

Both AEA and PEA, albeit at different concentrations, activate TRPV1 [53,62,63]. As previously reported, the BCCAO/R is followed by increased concentrations of AEA and PEA in the cortex and increased concentration of AEA in plasma [16]. Interestingly, in the spinal cord, AEA can modulate, in a concentration-dependent fashion, the TRPV1-positive fibers being excitatory through TRPV1 and inhibitory through CB1 [64]. In our experimental conditions, upon BCP treatment, the N-acyl amide changes induced by the BCCAO/R showed a different direction since the cerebral tissue AEA concentration was unchanged and PEA concentration decreased, while in the plasma AEA concentration decreased [18]. Altogether, this setting can denote why we observed that the TRPV1 protein levels were down-regulated following BCCAO/R in vehicle-treated rats while were upregulated upon BCCAO/R in BCP-treated rats. One should also consider that, due to the presence of a significant BCP × BCCAO/R interaction, the increase of TRPV1 may represent a dynamic adaptation contributing to modulating the endogenous pro-inflammatory milieu prompted by the BCCAO/R and tuned by BCP.

PEA is a known anti-inflammatory lipid mediator [65,66]. PEA has been shown to activate TRPV1 at high concentrations [67] and shares with other lipid mediators, such as the lipoperoxides, the ability to bind to and activate the intracellular receptor PPARalpha [68]. On the other hand, through the entourage effect [69], PEA might also interfere with AEA degradation and further enhance the activation of CB receptors and TRPV1 [53,67]. From a translational point of view, it is also important to underline that, conversely, both TRPV1 and PPARalpha are involved in regulating PEA-induced Ca++ signaling [70]. 

Though it does not represent the subject of this study, the interplay between both CB or PPARalpha receptors and TRPV1 could be an additional route to regulate the TRPV1 activation and desensitization [70]. A PPARalpha-dependent pathway through which PEA may activate TRPV1 has been demonstrated in cultured sensory neurons [70]. In the BCCAO/R model, as previously shown [18], increased expressions of CB1, CB2, and PPARalpha are among the effects of pre-treatment with BCP, further suggesting that TRPV1 may likely be modulated by the second messenger cascades activated by CB and PPARalpha receptors and concur with them in the response against the pro-inflammatory environment. 

### 3.2. TRPV1-like Immunoreactive Structures in BCCAO/R with and without BCP

The TRPV1 expression in the CNS has been investigated in numerous previous studies using different techniques; however, the available information concerning its cellular localization and regional distribution is still contradictory [48,49,71,72,73,74,75,76]. On the other hand, there is consensus regarding the involvement of TRPV1 in functions that, on the base of its multimodal activation, differ depending on the neuronal system/neuropathological condition considered [50,77]. 

Similar to available studies on the distribution of TRPV1-LI in the rodent CNS [49,50], our results show that TRPV1 occurs in the frontal cortex where it localizes to cell bodies, mainly with an intracytoplasmic distribution, and dot- and thread-like elements suggestive of nerve fibers. However, at variance with previous studies [49,50], in our hands, the cortical TRPV1-labeling was light (under baseline and BCP-treated BCCAO/R conditions) to virtually absent (in BCCAO/R condition after vehicle-treatment) and, when present, the TRPV1-positive structures often underlined the course of some blood vessels in the cortex. It remains to clarify whether the vessel-associated immunoreactive elements we observed are genuine nerve fibers, as suggested by their varicose appearance, or stand for non-nervous vascular elements as shown by transmission electron microscopy (TEM) in the rat brain [49]. TRPV1 mRNA has also been reported in several other vascular beds in the rat [76,78]. Interestingly, the TRPV1 on sensory nerve terminals mediates local vasodilation, while the vascular TRPV1 leads to vasoconstriction [76]. Cavanaugh and coll [76] suggested that, in thermoregulatory tissues, activation of TRPV1 on vascular smooth muscle cells could counteract nerve-related changes in vascular tone in response to physiological TRPV1 agonists. Provided that a diverging role for TRPV1 was also possible in the brain vessels, it could be speculated that the TRPV1 increase induced by BCP in BCCAO/R compared to sham rats is partly involved in controlling the homeostasis of the cortical vascular bed under the reperfusion challenge. 

This last inferring is further supported by our findings that after the BCP-treatment, beyond the TRPV1 increase, significant GFAP downregulation and partial TRPV1-/GFAP-like immunoreactivity colocalization were detectable in the BCCAO/R compared to the vehicle-treated rats. While the evaluation of GFAP expression in our setting warrants further investigations to clarify the astroglial involvement in the regulation of TRPV1 relative levels in the brain after BCCAO/R, it is tempting to hypothesize that the TRPV1-bearing astrocytes may be contributing to the regulation of vascular tone under the hypoperfusion/reperfusion challenge. Interestingly, TRPV1-positive astrocytes have already been demonstrated in the rat spinal cord [79] and brain [49]. As already suggested [49], astrocytes may represent another critical element by which TRPV1 may intervene to regulate the vascular bed, either by modulating the tone of cerebral vessels or modulating the blood-brain barrier permeability in response to an inflammatory challenge. Indeed, though with a time longer than that of our BCCAO/R model, it has been shown that the inflammatory milieu generated by 2.5 h of ischemic brain injury stimulated a strong astroglial response. This response was featured by cell hypertrophy and hyperplasia [77,80], and it was suggested that it may lead to an increased expression of GFAP, partly because the number of GFAP-positive astrocytes increases [81]. 

Concerning the microglial marker Iba1, the significant general effect of the BCCAO/R main factor evidenced by the ANOVA analysis was not paralleled by statistically significant *post-hoc* differences in Iba1 relative levels between sham and BCCAO/R rats, with or without BCP. The TRPV1/Iba1 double immunolabeling argues against the possibility of a microglial expression of the receptor, and no patent changes in the morphology of Iba1-positive elements could be either observed. The fact that, in our hands, no patent histological damage, as to induce evident microglia activation, could be detected in the rat brain after BCCAO/R is difficult to interpret. Numerous previous literature data report heavy microglial activation in the acute phase of an ischemic stroke in several experimental protocols [82], the shortest survival time possibly being that of 20 min after forebrain ischemia of 25 min [83]. However, more recent findings showed that some microglial cells were activated within 10 min following BCCAO and reached a strong reaction within 20 min without reperfusion [84] and refs therein. Interestingly, it has also been shown that the extent of microglial reaction not only does not increase with the extension of the survival time after BCCAO, serving possibly as a neuroprotectant during the vessel occlusion, but can easily be reversed by reperfusion [84]. This suggests the possibility that in our setting we are merely underestimating the microglia changes or searching too late for them to be appreciable.

### 3.3. BDNF and trkB in BCCAO/R without and after BCP

In the vehicle-treated rats, the ANOVA analysis of frontal cortex homogenates after 30 min of BCCAO followed by 60 min of reperfusion revealed no statistically significant change of BDNF relative protein levels. It has to be taken into account that though not resulting as statistically significant with the post-hoc test, as previously reported [20], there was a slight reduction of BDNF relative levels in the BCCAO/R rats compared to the sham animals. So far, no studies regarding the acute BCCAO/R-induced changes of mature BDNF protein in the brain cortex at times earlier than 2 h are available. Our findings appear consistent with an earlier study investigating the BDNF protein expression levels in the neocortex in a rat model of 10 min of BCCAO with hypotension followed by 2 h of reperfusion and reporting that low basal expression of BDNF protein further reduced at 24 h [85]. Interestingly, in their study Kokaia et al. [85] reported no changes in mRNA expression in the neocortex after the ischemia; this, besides pointing out the importance of analyzing BDNF at the protein level, also led to suggest that the cortex, having low baseline BDNF protein levels, is a region intrinsically vulnerable to ischemia and that BDNF, though decreasing during the 24 h, can play a neuroprotective role after the BCCAO/R challenge. To confirm this inferring, it could be relevant that in our experimental setting the trkB relative levels underwent a concomitant increase that may help assure adequate BDNF trophic signaling. Accordingly, upregulation of the trkB mRNA has been reported after severe ischemia [85,86,87,88].

Our results showed that the BCP administration, in both sham and BCCAO/R rats, had effects on BDNF, inducing a general decrement of its levels, and on trkB, causing an increment of its expression. It is tempting to speculate that the relative levels of trkB and its ligand complement each other to confer resistance and/or reduce brain damage following the BCCAO/R in vehicle-treated animals. Interestingly, after the BCP treatment, a marked increase of trkB relative levels occurred in the BCCAO/R as compared to sham rats, while BDNF was unchanged in the same conditions. This mismatch leads to envisaging alternative ways of trkB activation, other than the ligand-dependent one, possibly induced by the interaction of the trkB with products of the complex molecular network induced by BCP. In this line, evidence shows that CB1 receptors can couple and transactivate receptor tyrosine kinases and serine-threonine kinases, like ERK, in cultured murine neuronal cells [89]. Moreover, it has also been shown that CB1, upon activation of AEA and 2-AG, can couple to trkB and trigger the Src kinase-dependent trkB transactivation during morphogenesis of cortical interneurons [90]. Interestingly, the CB1-trkB coupling had been demonstrated not to involve the increases of BDNF synthesis and/or translation [90]. In another recent study, AEA and 2-AG, in a CB1- and TRPV1-dependent fashion, trigger a coordinated activation of trkB: the activation of CB1 being crucial for trkB coupling and transactivation in inhibitory cortical interneurons, while the activation of TRPV1 triggering the BDNF release and subsequent activation of trkB in pyramidal cells [91]. Since, using the same BCCAO/R model, we have previously shown that in the BCCAO/R rats BCP treatment induced the up-regulation of CB1, CB2, and PPARalpha [18], it is tempting to speculate that the increase of trkB levels in the BCP-treated BCCAO/R rats is subject to the CB1 coupling and transactivation. It is also relevant that BCP may reduce the ischemic injury in rat cortical neurons/glia mixed cultures subject to oxygen-glucose deprivation/re-oxygenation via CB2-induced activation of the AMPK/CREB pathway [92] and subsequent increased expression of the BDNF, a known CREB target gene product. 

Finally, in the complexity of the eCB system, it should also be considered that BDNF in association with eCBs may modulate neuronal activity and inhibit synaptic transmission through a BDNF-trkB-eCB signaling pathway [93]. Activation of trkB might then induce eCB release via the phospholipase C γ (PLCγ)/diacylglycerol lipase (DGL) pathway, retrogradely activating the presynaptic CB1 receptors [93,94]. The cross-talk between BDNF/trkB and eCB systems has been supported further by evidence that trkB activation, via intracellular calcium transients, increases eCB synthesis and release in the corticostriatal pathway [95].

### 3.4. Final Remarks

BCP safety has been evaluated for its use in medical food products and, given its extremely low toxicity (its acute oral lethal dose LD50 in rats being >5000 mg/kg body weight) [96,97,98], BCP has been approved recently as a food or cosmetic additive by the U.S. Food and Drug Administration (FDA) [99]. Numerous recent studies focusing on the formulation of different drug delivery systems aimed to optimize the oral bioavailability of BCP [97] and refs therein.

The present results add to previously published data [18] showing that BCP prevents or ameliorates the BCCAO/R-induced molecular dysregulation by reducing the lipoperoxidation (at both central and peripheral levels) and the COX-2 protein expression [18], activating the eCB system, and increasing the PPARalpha receptor expression [18]. Certainly, the BCP modulation of the TRPV1, BDNF, and trkB levels poses further questions concerning its modality of action given the cross-talk, possibly modulated by second messenger cascades, between both CB or PPARalpha receptors and TRPV1 [70], between CB1 and trkB [90], and between the BDNF/trkB and eCB systems [93,94,95]. 

## 4. Materials and Methods

### 4.1. Animals and Keeping

Adult male Wistar rats (Harlan, Udine, Italy), weighing 210 ± 20 g (mean ± SD), for 1 week before the experiment began were housed under controlled temperature (21 ± 2 °C), a 12 h light/dark cycle, and relative humidity (60 ± 5%), avoiding any distress of animals. Rat handling and care, and experimental procedures conformed with the guidelines of the Animal Ethics Committee of the University of Cagliari (approval code No. 06/2013, 05/31/2013), in compliance with national (D.L. n.116, Gazzetta Ufficiale della Repubblica Italiana, Additional 40, 18 February 1992, and subsequent modifications) and international laws and policies (EEC Council Directive 86/609, OJ L 358, 1, 12 December 1987). Standard laboratory food (A04, Safe, Augy, France) and water were freely available *ad libitum*.

Animals were not fed for 12 h before surgery. Rats (*n* = 44) were randomly assigned to two groups; 6 h before the surgery, rats were gavage fed a pre-treatment: one group (vehicle-treated) received the vehicle, i.e., 0.3 mL of sunflower oil, the other was given a dose of BCP (Sigma-Aldrich, St. Louis, MO, USA), i.e., 40 mg/per rat in 300 μL of sunflower oil (corresponding to 180 mg/kg) [18]. Each group was subdivided into sham-operated or submitted to BCCAO/R.

### 4.2. Surgery

The surgical procedure (adapted from the method of Iwasaki et al. [100]) was performed between 1:00 and 4:30 p.m. An intraperitoneal injection of Equithesin (16.2% *w/w* pentobarbital, 4.2% *w/v* chloral hydrate, 39.6% *w/w* propylene glycol, 2.12% *w/v* MgSO_4_, and 10% *w/w* ethanol in sterile distilled H_2_O) (5 mL/100 g body weight) was used as anesthesia. A midline incision of the neck was followed by blunt muscle dissection to expose the common carotid arteries (CCA), taking care to avoid hurting the vagus nerve. To reduce the cerebral blood flow, the CCA were clamped for 30 min with 2 atraumatic microvascular clips. The reperfusion period was attained by removing the clips and restoring blood flow for 60 min. The sham-operated rats underwent surgery without CCA occlusion and, for this reason, they represented the control animals, used to determine the effects of anesthesia and surgical manipulation on the results.

### 4.3. Sampling

At the end of the BCCAO/R procedure, brain samples were collected either as fresh tissue for Western blot or after fixation by transcardial perfusion with ice-cold fixative solution (0.1 M phosphate buffer (PB), pH 7.3, and 4% paraformaldehyde). For Western blot analysis, the cerebral cortex was rapidly dissected from the rest of the brain and transversely cut, respectively, at the approximate bregma level of −1.0 mm [101] for the frontal cortex, and of −4.5 mm for the temporal-occipital cortex, used as a control cortical area not supplied by the internal carotid artery branches; specimens were then frozen at −80 °C until analysis. For immunohistochemical assays, brains were post-fixed by immersion in freshly prepared fixative, pH 7.3, for 4–6 h at 4 °C, and then rinsed in 0.1 M PB, pH 7.3, containing 20% sucrose. For both Western blot and immunohistochemical assays, the investigator was blind to the experimental condition of rats.

### 4.4. Western Blot

Tissue homogenates (*n* = 24 vehicle-treated and 20 BCP-treated rats were prepared in a 2% solution of sodium dodecyl sulfate (SDS) containing a cocktail of protease inhibitors (cOmplete, Mini Protease Inhibitor Cocktail Tablets, Roche, Basel, Switzerland). Protein concentrations were determined according to Lowry’s protein assay method [102], using bovine serum albumin as the standard. Proteins for each tissue homogenate (40 μg) were diluted 3:1 in 4× loading buffer (NuPAGE LDS Sample Buffer 4×, Novex by Life Technologies, Carlsbad, CA, USA), heated to 95 °C for 7 min, and separated by SDS-polyacrylamide gel electrophoresis (SDS-PAGE) using precast gradient gel (NuPAGE 4–12% Bis-Tris Gel Midi, Novex by Life Technologies) in the XCell4 Sure Lock^TM^ Midi-Cell chamber (Life Technologies). Internal molecular weight (mw) standards (Precision Plus Protein^TM^ WesternC^TM^ Standards, Bio-Rad, Hercules, CA, USA) were run in parallel. Two gels at a time were run for immunoblotting and Coomassie blue staining, respectively. Proteins for immunoblotting were transferred by electrophoresis on a polyvinylidene fluoride synthetic membrane (Amersham Hybond^TM^-P, GE Healthcare, Little Chalfont, UK) using the Criterion^TM^ Blotter (Bio-Rad). Blots were blocked by immersion in 20 mM Tris base and 137 mM sodium chloride (TBS), containing 0.1% Tween 20 (TBS/T) and 5% milk powder, for 60 min, at room temperature (RT). The polyclonal antibodies raised in rabbit against TRPV1 (AbCam, Cambridge, UK), diluted 1:1000, BDNF (Cat# R-017, Biosensis Pty Ltd., Thebarton, Australia), diluted 1:2000, trkB (Cat# 794 sc-12, SCBT), diluted 1:1000, GFAP Cat# Z0334, Dako, Glostrup, Denmark), diluted 1:2000, and Iba1(Cat#019-19741, Waco Pure Chemical Industries, Ltd., Osaka, Japan), diluted 1:1000 in TBS containing 5% dried milk and 0.02% sodium azide (Sigma, Milan, Italy) were used as primary antisera. Incubations were performed at 4 °C and lasted 48 h. Controls for equal-loading of the wells were obtained by immunostaining the membranes, as above, for glyceraldehyde-3-phosphate dehydrogenase (GAPDH), using a mouse monoclonal anti-GAPDH antibody (MAB374, EMD Millipore, Darmstadt, Germany), diluted 1:1000, as the primary antiserum. After rinsing in TBS/T, blots were incubated at RT, for 60 min, with peroxidase-conjugated goat anti-rabbit (Cat#9169, Sigma Aldrich, St. Louis, MO, USA), diluted 1:10,000, and anti-mouse serum (AP124P, EMD Millipore, Darmstadt, Germany), diluted 1:5000 in TBS/T, as the secondary antiserum. To control for non-specific staining, blots were stripped and incubated with the relevant secondary antiserum. Moreover, to test out antibody specificity and cross-reactivity, the anti-BDNF antibody was preabsorbed with 200 ng of rhBDNF (Cat# B-257, Alomone Labs, Jerusalem, Israel). Membranes were then rinsed in TBS/T, and protein bands were revealed with the Western Lightning Plus ECL (Cat# 103001EA, PerkinElmer, Waltham, MA, USA), according to the manufacturer’s protocol, and visualized using the ImageQuant^TM^ LAS-4000 (GE Healthcare, Little Chalfont, UK). Approximate molecular weight (mw) and relative optical density (O.D.) of the immunostained protein bands were evaluated by a blinded examiner. Quantification of O.D. of each sample was performed using the Image Studio Lite Software (LI-COR Biosciences-GmbH). The ratios between the intensity of the bands immunolabelled for TRPV1, BDNF, trkB, GFAP, Iba1, and those positive for GAPDH ones were used to compare the relative levels of protein expression in the examined experimental conditions, and is shown as graphs in Figure 1.

### 4.5. Immunohistochemistry 

Coronal serial sections (14 μm thick) of frontal and temporal-occipital cortex were cut with a cryostat, collected on chrome alum-gelatin-coated slide, and processed by the avidin-biotin-peroxidase complex (ABC) and the indirect immunofluorescence (IIF) techniques. Brains of vehicle (*n* = 12)—and BCP—treated (*n* = 10) rats were processed in pairs on the same slide.

For the ABC, slides were firstly treated with 0.1% phenylhydrazine in phosphate-buffered saline (PBS) containing 0.2% Triton X-100 (PBS/T) to inhibit the endogenous peroxidase activity, then with 20% of normal goat serum (Cat# S-1000, Vector Labs Inc., Burlingame, CA, USA) for 1 h at RT. Rabbit polyclonal antibodies against TRPV1 (Cat# ab10296, AbCam, Cambridge, UK), diluted 1:600, BDNF (Cat# R-017, Biosensis Pty Ltd., Thebarton, Australia), diluted 1:400, and trkB (SCBT, Santa Cruz, CA, USA), diluted 1:500, were used as primary antibody. A biotin-conjugated goat anti-rabbit serum (Cat# BA-1000, Vector Labs Inc., Burlingame, CA, USA), diluted 1:400, was used as the secondary antiserum. The reaction product was revealed with the ABC (Cat#G011-61, BioSpa Div., Milan, Italy), diluted 1:250, followed by development with the chromogen solution of 0.05% 3,3′-diaminobenzidine (Sigma Aldrich, St. Louis, MO, USA), 0.04% nickel ammonium sulfate and 0.01% hydrogen peroxide in 0.1 M PB, pH 7.3.

For the IIF, polyclonal antibodies raised in goat against TRPV1 (Cat# sc-12498, SCBT, Santa Cruz, CA, USA), diluted 1:100, and in rabbit against BDNF (Biosensis Pty Ltd., Thebarton, Australia), diluted 1:100, trkB (SCBT, Santa Cruz, CA, USA), diluted 1:100, GFAP (Cat# Z0334, Dako, Glostrup, Denmark), diluted 1:500, and Iba1 (Cat#019-19741, Waco Pure Chemical Industries, Ltd., Osaka, Japan), diluted 1:1000 were used as primary antiserum and incubations were run overnight at 4 °C. Alexa Fluor 488- or Alexa Fluor 594-conjugated donkey antisera against goat and rabbit proteins (Invitrogen, Eugene, OR, USA), diluted 1:500, were used as the secondary antiserum. All antisera and the ABC were diluted in PBS/T. Incubation with primary antibodies was carried out overnight at 4 °C. Incubations with secondary antiserum and ABC lasted 60 min and 40 min, respectively, and were performed at RT. Negative control preparations were obtained by incubating tissue sections in parallel with either PBS/T, alone, or (i) with the relevant primary antiserum pre-absorbed with an excess of the corresponding peptide antigen [for the anti-BDNF: rhBDNF (Cat# B-257, Alomone Labs, Jerusalem, Israel); for the trkB: Cat# sc-12 P (SCBT, Santa Cruz, CA, USA)]; (ii) or by the omission of the primary antibody; or (iii) by substituting the primary antibody with normal serum. Slides were observed with an Olympus BX61 microscope and digital images were acquired with a Leica DFC450 C camera.

### 4.6. Statistical Analysis

Statistical evaluation was performed on densitometric data obtained with the Western blot assays. Data from the four experimental conditions, i.e., vehicle-treated- or BCP-treated sham animals, vehicle-treated and -or BCP-treated BCCAO/R rats, are depicted in Figure 1 as the mean ± standard error of the mean (S.E.M.). Two-way analysis of variance (ANOVA) (main factors: (a) BCP-treatment (i.e., vehicle- vs. BCP-treatment) and (b) BCCAO/R (i.e., sham-operation vs. BCCAO/R) was performed using Prism 6.01 for Windows (GraphPad Software, La Jolla CA, USA). Wherever appropriate (i.e., *p* for the main factors and their interaction <0.05), multiple pair-wise contrasts were made, and the multiplicity adjusted *p*-value for each comparison was calculated using Tukey’s *post hoc* test. 

## 5. Conclusions

To the best of our knowledge, the present study is the first to show that the BCP pretreatment has modulatory effects on the ionotropic cannabinoid receptor TRPV1, the BDNF/trkB system, and the glial marker GFAP. Collating the evidence we obtained so far regarding the BCP properties [18], the present data support the concept that BCP may activate multiple mechanisms to cope with the BCCAO/R-induced molecular dysregulation and indicate that it may be an excellent therapeutic agent to preserve the tissue from the pathophysiological sequelae of the hypoperfusion/reperfusion challenge.

## Figures and Tables

**Figure 1 ijms-23-03633-f001:**
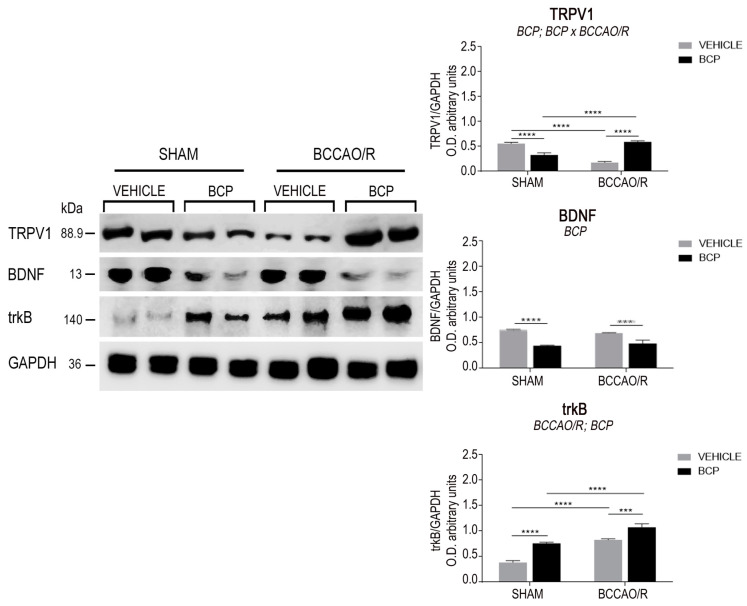
Western blot analysis of the frontal cortex homogenates in vehicle-treated and beta-caryophyllene (BCP)-treated rats either in sham-operated or after bilateral common carotid artery occlusion followed by reperfusion (BCCAO/R). (**Left**) Transient receptor potential vanilloid type-1 receptor (TRPV1), brain-derived neurotrophic factor (BDNF), tyrosine kinase receptor B (trkB), glyceraldehyde-3-phosphate dehydrogenase (GAPDH) proteins. (**Right**) Densitometric analysis of the band gray levels expressed as a percentage of the optical density (O.D.) ratio of TRPV1-, BDNF-, trkB-positive bands to those immunostained for GAPDH. Data are reported as the mean values of 12 vehicle-treated and 10 BCP-treated rats for each experimental condition. Error bars depict the standard error of the mean (S.E.M.). Asterisks denote significant differences. Two-way ANOVA with the Tukey’s test for post-hoc analyses was applied to evaluate statistical differences between groups. BCCAO/R, significant effect of BCCAO/R; BCP, significant effect of BCP-treatment; BCP × BCCAO/R, significant BCP-treatment × BCCAO/R interaction. *** *p* < 0.001; **** *p* ≤ 0.0001 (see Table 1 for *F*-values and *p*-values relevant to effects of BCCAO/R and BCP pre-treatment and to the interaction between them).

**Figure 2 ijms-23-03633-f002:**
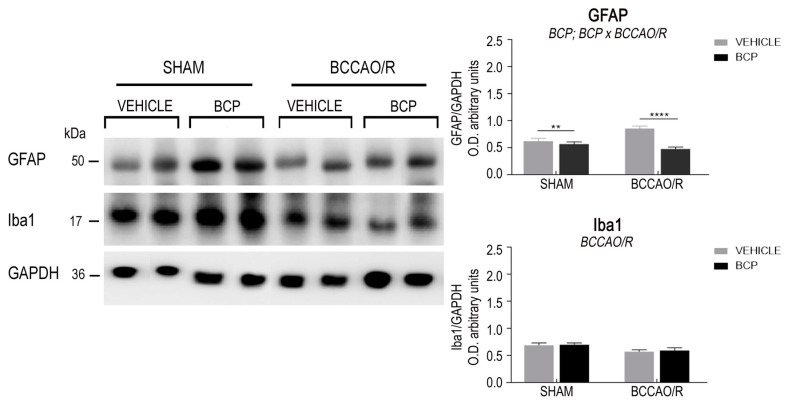
Western blot analysis of the frontal cortex homogenates in vehicle-treated and beta-caryophyllene (BCP)-treated rats either in sham-operated or after bilateral common carotid artery occlusion followed by reperfusion (BCCAO/R). (**Left**) Glial fibrillary acidic protein (GFAP), ionized calcium-binding adapter molecule 1 (Iba1), glyceraldehyde-3-phosphate dehydrogenase (GAPDH) proteins. (**Right**) Densitometric analysis of the band gray levels expressed as a percentage of the optical density (O.D.) ratio of GFAP- and Iba1-positive bands to those immunostained for GAPDH. Data are reported as the mean values of 12 vehicle-treated and 10 BCP-treated rats for each experimental condition. Error bars depict the standard error of the mean (S.E.M.). Asterisks denote significant differences. Two-way ANOVA with the Tukey’s test for post-hoc analyses was applied to evaluate statistical differences between groups. *BCCAO/R*, significant effect of BCCAO/R; *BCP*, significant effect of BCP-treatment; *BCP* × *BCCAO/R*, significant BCP-treatment × BCCAO/R interaction. ** *p* < 0.01; **** *p* ≤ 0.0001 (see Table 1 for *F*-values and *p*-values relevant to effects of BCCAO/R and BCP pre-treatment and to the interaction between them).

**Figure 3 ijms-23-03633-f003:**
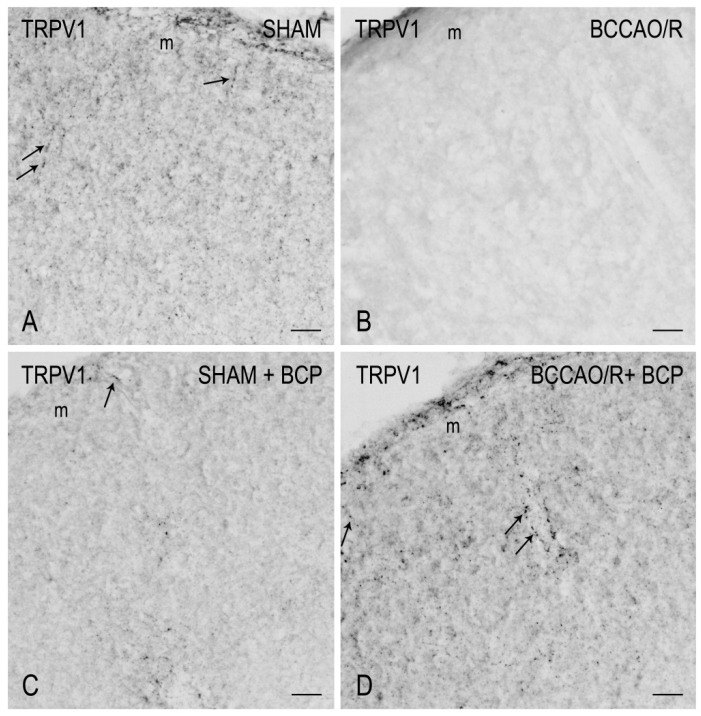
Transient receptor potential vanilloid type-1 receptor (TRPV1)-like immunoreactivity in coronal sections of frontal cortex of sham-operated and bilateral common carotid artery occlusion followed by reperfusion (BCCAO/R) rats, pre-treated with either the vehicle alone (**A**,**C**) or with beta-caryophyllene (BCP) (**B**,**D**). TRPV1-immunoreactive neural processes and punctate elements are distributed across the whole thickness of the cortex, often associated with blood vessels (arrows). Panels are representative of observations carried out in 6 rats for each group. m, molecular layer. Scale bars: 25 μm.

**Figure 4 ijms-23-03633-f004:**
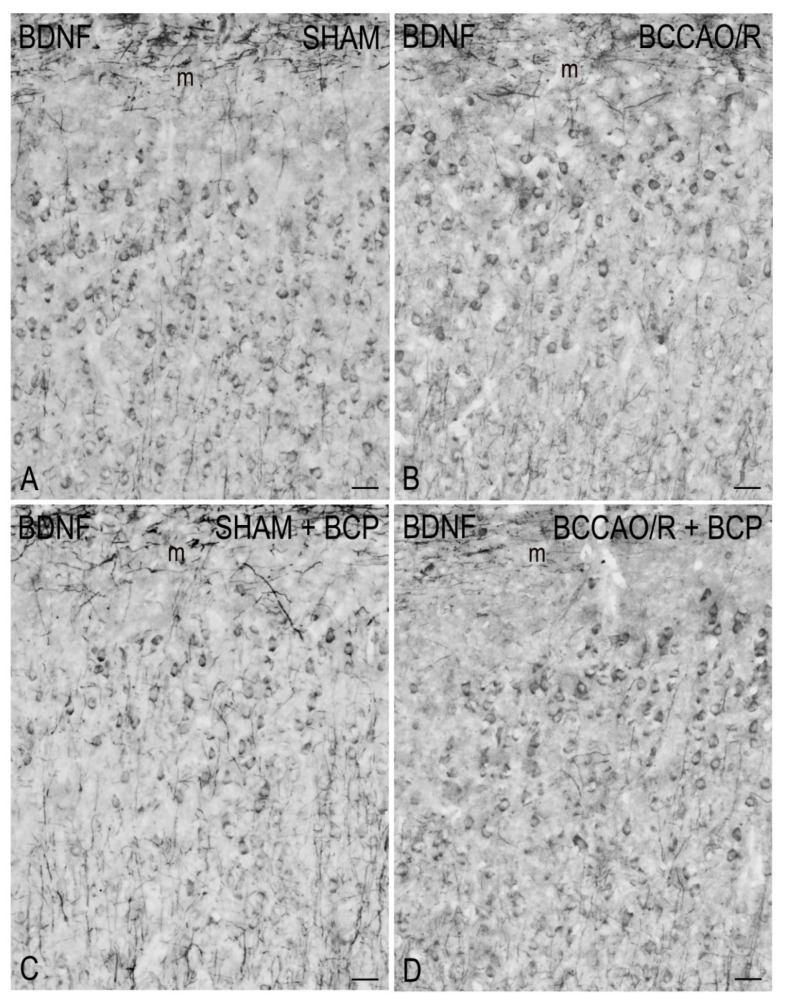
Brain-derived neurotrophic factor (BDNF)-like immunoreactivity in coronal sections of frontal cortex of sham-operated and bilateral common carotid artery occlusion followed by reperfusion (BCCAO/R) rats, pre-treated with either the vehicle alone (**A**,**C**) or with beta-caryophyllene (BCP) (**B**,**D**). Immunoreactive neuronal perikarya, neuronal processes, and punctate elements are distributed across the whole thickness of the cortex. Arrows point to positive neuronal perikarya. Panels are representative of observations carried out in 6 rats for each group. m, molecular layer. Scale bars: 25 μm.

**Figure 5 ijms-23-03633-f005:**
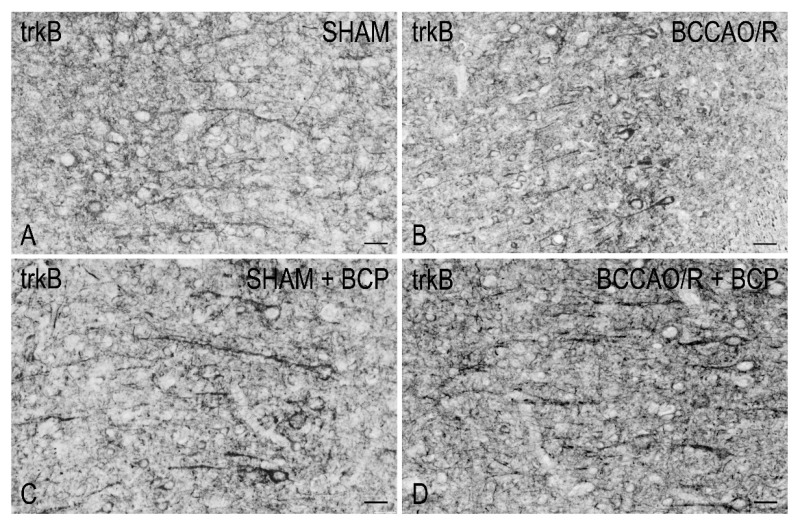
Tyrosine kinase receptor B (trkB)-like immunoreactivity in coronal sections of frontal cortex of sham-operated and bilateral common carotid artery occlusion followed by reperfusion (BCCAO/R) rats pre-treated with either the vehicle alone (**A**,**C**) or with resveratrol (BCP) (**B**,**D**). Positive neuronal processes, punctate elements, and neuronal perikarya are distributed across the whole thickness of the cortex. Arrows point to immunostained neuronal perikarya. Panels are representative of observations carried out in 6 rats for each group. Scale bars:  25 μm.

**Figure 6 ijms-23-03633-f006:**
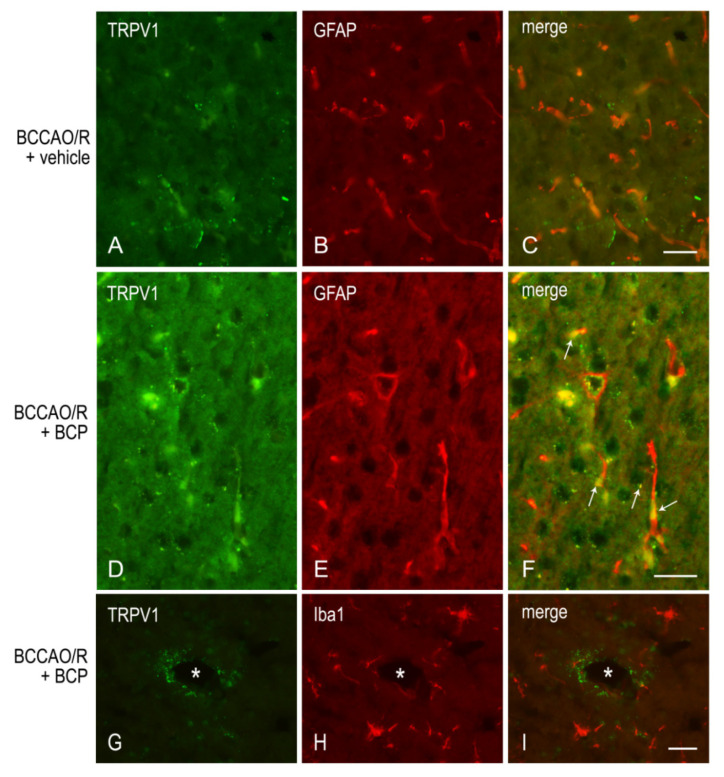
Double immunofluorescence for TRPV1 (**A**,**C**,**D**,**F**,**G**,**I**) and either glial fibrillary acidic protein (GFAP) (**B**,**C**,**E**,**F**) or Iba1 (**H**,**I**) in representative coronal sections of frontal cortex of BCCAO/R + vehicle (**A**–**C**) and BCCAO/R + BCP rats (**D**–**I**); arrows point to TRPV1/GFAP double-labelled elements. *, blood vessel. Scale bars: A, B = C: 25 μm; D, E = F: 25 μm; G, H = I: 25 µm.

**Table 1 ijms-23-03633-t001:** *F*-values and significance levels from two-way ANOVA performed on mean values of marker protein levels obtained after transient bilateral common carotid artery occlusion followed by reperfusion (BCCAO/R) and beta-caryophyllene (BCP) pre-treatment by means of Western blot in the rat frontal cortex.

Marker	ANOVA Factors						
	BCCAO/R		BCP Treatment		BCP Treatment × BCCAO/R		
	*F*-Value	*p*-Value	*F*-Value	*p*-Value	*F*-Value	*p*-Value	*df*
TRPV1	3.259	ns	8.031	0.0072	83.05	<0.0001	1, 40
BDNF	0.1266	ns	72.82	<0.0001	3.224	ns	1, 40
trkB	61.64	<0.0001	38.95	<0.0001	0.8211	ns	1, 40
GFAP	2.669	ns	26.34	<0.0001	14.05	0.0006	1, 40
Iba1	7.871	0.0007	0.05863	ns	0.1311	ns	1, 40

BDNF, brain-derived neurotrophic factor; GFAP, glial fibrillary acidic protein; Iba1, ionized calcium-binding adapter molecule 1; trkB, tyrosine kinase receptor B; TRPV1, transient receptor potential vanilloid type-1 receptor. *df*, degrees of freedom; ns, not significant.

## Data Availability

The data presented in the current study are available from the corresponding author on reasonable request.

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
