# Peer review of "Anti-Inflammatory Effect of Beta-Caryophyllene Mediated by the Involvement of TRPV1, BDNF and trkB in the Rat Cerebral Cortex after Hypoperfusion/Reperfusion"

_ijms, 2022, doi:10.3390/ijms23073633_

Round 1
Reviewer 1 Report
The manuscript titled "Anti-inflammatory effect of beta-caryophyllene mediated ..." by Maria Pin Serra is quite a good work that deserves to be published in the IJMS.
The work, apart from a slight lack of diligence in preparation, does not contain too many errors. From the substantive side, it's hard to fault anything.
Below is a short list of my comments / suggestions.
1. Title - I would change it a bit - probably too many abbreviations in it.
2. Abstract - please rewrite the last two sentences - because they sound somehow strange.
3. Are there too many keywords? Besides, some of them are contained within others, so why produce so many of them there?
4. The introduction is fine, but the authors have overdone the last paragraph a bit. Please emphasize the purpose of the manuscript.
5. Please fix Fig 1 somehow - it is unreadable - especially those little panels.
6. Same with Fig 2.
7. Is the last sentence in the summary necessary?
The rest of the work is really fun and nice to read.
Please correct the work and send it back - I don't have to see it anymore.
Author Response
-
- Title - I would change it a bit - probably too many abbreviations in it.
Re: The authors appreciate the referee’s suggestion and revised the Title according to it, as follows "Anti-inflammatory effect of beta-caryophyllene mediated by the involvement of TRPV1, BDNF and trkB in the rat cerebral cortex after hypoperfusion/reperfusion".
- Abstract - please rewrite the last two sentences - because they sound somehow strange.
Re: We understand the reviewer's point and apologize for the lack of clarity. To meet the reviewer comments we rewrote the sentences "After BCP-treatment…. support in the changes triggered by the BCCAO/R." that now read "The BCP-treatment caused a decrease of BDNF and an increase of trkB levels in both sham and BCCAO/R conditions, while induced opposite changes in the case of TRPV1, whose levels became higher in BCCAO/R and lower in sham conditions. Present results highlight the role of BCP in modulating early events of the cerebral inflammation triggered by the BCCAO/R through the regulation of TRPV1 and the BDNF-trkB system."
- Are there too many keywords? Besides, some of them are contained within others, so why produce so many of them there?
Re: We understand the referee’s point and thank you for the suggestion since it allowed us to modify the keywords and avoid redundancy. The terms "cerebral hypoperfusion/reperfusion; brain cortex; biomarkers" have been removed and now, according also to the journal instructions allowing to list three to ten keywords, the list now is as follows: beta-caryophyllene, neuroinflammation; acute bilateral-common carotid artery occlusion; TRPV1; BDNF/trkB; GFAP; Iba1; Western blot; immunohistochemistry.
- The introduction is fine, but the authors have overdone the last paragraph a bit. Please emphasize the purpose of the manuscript.
Re: We thank the reviewer for the suggestion. The last paragraph has been revised thoroughly and slightly shortened to emphasize the aim of the study. Sentences now read:
Therefore, given that revascularization of common carotid arteries is likely to induce a pro-inflammatory milieu at the end of the hypoperfusion process, here we extend our previous observations on the neuropreventive effects of an acute dose of BCP on the molecular changes induced by the BCCAO/R in the rat. With this aim, utilizing Western blot and immunohistochemical assays, we examined the levels of TRPV1, BDNF, and its high affinity tropomyosin receptor kinase B (trkB), as well as those of the glial fibrillary acidic protein (GFAP) and ionized calcium-binding adapter molecule 1 (Iba1). The investigated regions were the frontal cortex, the brain region that is affected by the hypoperfusion/reperfusion due to the transient occlusion of the common carotid arteries [16], and the temporal-occipital cortex, an area supposedly not influenced by the BCCAO/R [16]. Results are discussed considering the possibility to use the phytocannabinoid BCP as a dietary supplement to prevent the changes in levels of TRPV1, BDNF, and trkB, taken as putative molecular markers of the adaptive tissue plasticity in response to the pro-inflammatory state induced by the cerebral hypoperfusion/reperfusion.
- Please fix Fig 1 somehow - it is unreadable - especially those little panels.
- Same with Fig 2.
Re: Authors thank the reviewer for noticing this fault. Hoping that a possibly flawed high-resolution view might have represented the cause of the unreadability, to meet the reviewer's point authors have improved the figures by enlarging the lettering font size. Revised Figures 1 and 1 are resubmitted consequently.
- Is the last sentence in the summary necessary?
Re: authors believe that this reviewer's point is contained within his/her second specific comment and thus kindly ask to refer to the relevant response above.
Reviewer 2 Report
In general, this work is really interesting and well written. My minor comments aim to increase the scientific soundness and clarity of it.
Line 42 – why “Bilateral Common Carotid Artery Occlusion followed by Reperfusion” is written in capital letters?
Line 66 – please expand the abbreviations “2-AG, AEA, PEA and OEA”
Line 104 – „trkB” stands for tropomyosin receptor kinase B. This abbreviation should be alos expanded.
Line 529, and throughout the text. – Immunofluorescence should be abbreviated to IF not IIF
Line 526 – in fact, in case of IHC the authors used species specific primary and secondary antibodies/sera not a single antibody/serum.
Line 182 – the authors stated that “BDNF- and trkB-like immunoreactive structures were quite abundant”. They also used words “frequently”, “occasionally” etc. But to judge that they needed any kind of reference scale. So what were the criteria for such quantification ? how many person were involved in counting procedures? Did they average somehow the obtained results ?
Line 572 – my majors concern is the Conclusion format. As is it rather resambles disscucion than conclusions. I would suggest authors to provide strong clear cut conclusions.
Author Response
Line 42 – why “Bilateral Common Carotid Artery Occlusion followed by Reperfusion” is written in capital letters?
Re: we thank the reviewer for noticing this aspect and apologize for the stylistic mistake. The capital letters have been changed to normal characters.
Line 66 – please expand the abbreviations “2-AG, AEA, PEA and OEA”
Re: we apologize for the stylistic mistake. Abbreviations have been justified with the extended names.
Line 104 – „trkB” stands for tropomyosin receptor kinase B. This abbreviation should be alos expanded.
Re: Abbreviation has been giustified with the extended name of the receptor.
Line 529, and throughout the text. – Immunofluorescence should be abbreviated to IF not IIF
Line 526 – in fact, in case of IHC the authors used species specific primary and secondary antibodies/sera not a single antibody/serum.
Re: Authors evaluated to keep the acronym IIF since it stands for "indirect immunofluorescence". As correctly pointed out by the reviewer, a fluorochrome-conjugated secondary antibody was used to visualize the complex antigen-primary antibody and not, as in the direct immunofluorescence, a fluorochrome-conjugated primary antibody.
Line 182 – the authors stated that “BDNF- and trkB-like immunoreactive structures were quite abundant”. They also used words “frequently”, “occasionally” etc. But to judge that they needed any kind of reference scale. So what were the criteria for such quantification ? how many person were involved in counting procedures? Did they average somehow the obtained results?
Re: Authors are grateful to the referee’s request since it allows to improve both clarity and the description of Methods and Results. In fact, the authors caught that it is not clear from the present phrasing that densitometry has been performed only on immunostained blots, whereas the immunohistochemical stainings have just been used to confirm the western blot protein detection with tissue localization of the selected markers.
To meet this specific comment, and for the sake of clarity, authors, therefore, specified that the densitometry was performed only on western blot data and the following changes have been made:
- paragraph Statistical analysis: the sentence "Statistical evaluation was performed on densitometric data obtained with the Western blot assays." has been added.
- paragraph Results-Immunohistochemistry:
- the sentence "By contrast with the scarcity of TRPV1-LI, the" has been added;
- the words "quite abundant" have been substituted by "numerous";
- the phrase "that, somewhat more frequently in BCP- than in vehicle-treated ones, occasionally ..." has been changed to ".., in both BCP- and vehicle-treated rats, rare..".
Line 572 – my majors concern is the Conclusion format. As is it rather resambles disscucion than conclusions. I would suggest authors to provide strong clear cut conclusions.
Re: We thank the reviewer for the suggestion that is definitely valid. To meet this point authors have revised the Conclusion paragraph that now reads as follows:
"To the best of our knowledge, the present study is the first to show that the BCP pretreatment has modulatory effects on the ionotropic cannabinoid receptor TRPV1, the BDNF/trkB system, and the glial marker GFAP. Collating the evidence we obtained so far regarding the BCP properties [18], the present data support the concept that BCP may activate multiple mechanisms to cope with the BCCAO/R-induced molecular dysregulation and indicate that it may be an excellent therapeutic agent to preserve the tissue from the pathophysiological sequelae of the hypoperfusion/reperfusion challenge."
Moreover, in order to give information regarding the BCP features and their effects on the BCCAO/R outcome, a brief text resuming them has been added within a new paragraph entitled "Final remarks" at the end of the "Discussion".